# Diminished measles immunity after paediatric liver transplantation—A retrospective, single-centre, cross-sectional analysis

Tobias Laue[1]*, Norman Junge[1], Christoph Leiskau[2], Frauke Mutschler[1], Johanna Ohlendorf[1], Ulrich Baumann[1]

1 Division for Paediatric Gastroenterology and Hepatology, Department of Paediatric Kidney, Liver, and Metabolic Diseases, Hannover Medical School, Hannover, Germany, 2 Paediatric Gastroenterology, Department of Paediatrics and Adolescent Medicine, University Medical Centre Goettingen, Georg August University Goettingen, Goettingen, Germany

* laue.tobias@mh-hannover.de

**Data Availability Statement:** All database files are available from the RepoMed database from Hannover Medical School (https://doi.org/10.

## Abstract

Liver transplantation in childhood has an excellent long-term outcome, but is associated with a long-term risk of infection. Measles is a vaccine-preventable infection, with case series describing severe courses with graft rejection, mechanical ventilation and even death in liver transplant recipients. Since about 30% of liver transplanted children receive liver transplants in their first year of life, not all have reached the recommended age for live vaccinations. On the contrary, live vaccines are contraindicated after transplantation. In addition, vaccination response is poorer in individuals with liver disease compared to healthy children. This retrospective, single-centre, cross-sectional study examines measles immunity in paediatric liver transplant recipients before and after transplantation. Vaccination records of 239 patients, followed up at Hannover Medical School between January 2021 and December 2022 were analysed. Twenty eight children were excluded due to stem cell transplantation, regular immunoglobulin substitution or measles vaccination after transplantation. More than 55% of all 211 children analysed and 75% of all those vaccinated at least once are measles seropositive after transplantation—48% after one and 84% after two vaccinations—which is less than in healthy individuals. Interestingly, 26% of unvaccinated children also showed measles antibodies and about 5–15% of vaccinated patients who were seronegative at the time of transplantation were seropositive afterwards, both possibly through infection. In multivariable Cox proportional hazards regression, the number of vaccinations (HR 4.30 [95% CI 2.09–8.83], p<0.001), seropositivity before transplantation (HR 2.38 [95% CI 1.07–5.30], p = 0.034) and higher age at time of first vaccination (HR 11.5 [95% CI 6.92–19.1], p<0.001) are independently associated with measles immunity after transplantation. In contrast, older age at testing is inversely associated (HR 0.09 [95% CI 0.06–0.15], p<0.001), indicating a loss of immunity. Vaccination in the first year of life does not pose a risk of non-immunity. The underlying liver disease influences the level of measles titres of twice-vaccinated patients; those with acute liver failure being the lowest compared to children with metabolic disease. In summary, vaccine response is poorer in children with liver disease. Liver transplant candidates should be vaccinated before transplantation even if this is earlier in the first

26068/mhhrpm/20231107-004). They have also been deposited on FigShare (https://doi.org/10.6084/m9.figshare.24996557).

**Funding:** The authors received no specific funding for this work.

**Competing interests:** I have read the journal's policy and the authors of this manuscript have the following competing interests: Prof. U. Baumann is a consultant for Albireo Pharma, Mirum Pharma, Alexion Pharma, Vivet Pharma and Nestlé. Dr. C. Leiskau has received payments for a presentation by Albireo pharma. The competing interests of Prof. Baumann and Dr. Leiskau do not alter our adherence to PLOS ONE policies on sharing data and materials. None of the above mentioned companies took part in study design, data collection and analysis, decision to publish, or preparation of the manuscript.

year of life. Checking measles IgG and re-vaccinating seronegative patients may help to achieve immunity after transplantation.

## Introduction

In recent decades, paediatric liver transplantation has become a well-established method of treatment for both acute and chronic liver disease [1–3]. Overall five-year survival rates are above 85% and up to 97% for those surviving the first year following transplantation [4]. However, patients have a significantly increased risk of infection both in the short and long term [5, 6], and is still the leading cause of mortality after transplantation in more than 4% of all cases [4]. Approximately 16% of paediatric organ recipients develop an infection in the first 5 years after solid organ transplantation (SOT) that could have been prevented by vaccination, resulting in increased morbidity, mortality and costs [7]. Measles is such a highly contagious vaccine-preventable infection (VPI): A case report mentions agranulocytosis and thrombocytopenic purpura after measles infection in a 2-year-old girl after living-related liver transplantation [8]. Another case report describes a liver transplant rejection in a 31-year-old due to a measles infection [9]. A case series describes five children after liver transplantation with pneumonia and laryngitis, but also more severe courses, with the need for mechanical ventilation, and even death [10].

In recent years, however, measles outbreaks have been frequently reported in immunocompetent individuals worldwide [11–13]. As a result, the recommended age for the first MMR vaccination has been lowered in several European countries and is now between 9 and 18 months [14, 15]. In particular, vaccine hesitancy and associated under-vaccination present a high risk for infection [13]. This also particularly threatens those children with liver disease: nearly 30% receive an organ in the first year of life [2, 6] and do not reach the recommended age for live vaccination, which is from 11 months in Germany [16] and 12 months in the USA [17]. Moreover, only 89% of U.S. children [18] and 81% of European children with chronic liver disease (observational study of the European Reference Network *TransplantChild*) [19] are age-appropriately immunized against measles at the time of liver transplantation. Until recently, there was no recommendation for live vaccination after SOT, as sufficient immune response may not be achieved due to immunosuppression [20]. However, studies show good immunogenicity and tolerability of the measles, mumps and rubella (MMR) vaccine even after liver transplantation [21, 22], so that since 2019 the American society of transplantation suggests they "may be administered in specific populations" [23].

Furthermore, studies suggest that the vaccination response of children with liver disease is impaired [19, 24, 25]. With regard to measles, Wu et al. demonstrated a significantly lower seroconversion after two doses of vaccination in only 42 of 50 (84%) children with biliary atresia compared to 96.7% in healthy individuals [26]. In addition, data suggest a decrease in seroprevalence over time following liver transplantation [27]. Only 46.9% of children were measles-seropositive after transplantation–however, 44 of 49 children received only one dose prior to transplant and almost 96% received a living-related liver transplantation [28]. Yoeli et al. analysed the post-transplant immunity against measles in paediatric liver transplant recipients: 16 of 72 (22.2%) children who received at least one vaccine dose prior to transplantation were non-immune [29]. However, there is little data on long-term immunity in patients after paediatric liver transplantation.

The aims of this retrospective cross-sectional study are to investigate measles immunity at least one year after paediatric liver transplantation in relation to the number of vaccine doses

before transplantation, as well as the underlying liver disease, in order to identify risk factors for non-immunity.

## Materials and methods

### Patients

In this retrospective, single-centre, cross-sectional study, data from children who underwent liver transplantation at Hannover Medical School (Germany) and were followed up between January 2021 and December 2022 was examined. Medical charts were reviewed to identify patient characteristics. Only patients with a certified vaccination record who were more than 10 months post-transplant were included in this study. If a child was re-transplanted more than 8 weeks after the first liver transplantation, only the data up to re-transplantation was used. Exclusion criteria were measles vaccination after transplantation, administration of rituximab (e.g. in the context of a lymphoproliferative post-transplantation disorder, PTLD), a haematopoietic stem cell transplant, or the regular administration of immunoglobulins. To exclude false-positive results, the interval between the measles IgG measurement and transfusions (red cell concentrates, platelet concentrate and fresh frozen plasma) had to be at least 12 weeks. Standard immunosuppression includes cyclosporine A or tacrolimus. Targeted drug trough levels after the first year following transplantation are 30–60 μg/L for cyclosporine A and 2–6 μg/L for tacrolimus. Prednisolone or mycophenolate mofetil (MMF) were added in patients with the need for intensified immunosuppression. Degree of rejection was assessed by the Rejection Activity Index (RAI score) according to the Banff Schema [30].

All parents/guardians of patients included in this study provided written informed consent at the time of hospital admission, allowing for the child's data to be used for scientific purposes. Patient data was anonymized prior to analysis. The study was conducted according to the guidelines of the Declaration of Helsinki and was approved by the local Ethics Committee of Hannover Medical School on 6th August 2021. The written approval document has the no. 9928_BO_K_2021.

### Antibody assessment and definition of measles immunity

Until the end of 2014, an enzyme-linked immunosorbent assay (measles IgG) with tests from Behring/Siemens Health Care Diagnostics was used in the Evolyzer from Tecan according to the manufacturer's instructions [31]. Results above 330 IU/ml were considered positive and correspond to serological immunity, all other titres were considered as negative and therefore stated as non-immune.

Since January 2015 measles IgG was measured using the Liaison indirect chemiluminescence immunoassay on the DiaSorin Liaison XL automated analyser according to the manufacturer's instructions [32]. The lower limit of detection (LOD) is 5 arbitrary units (AU)/ml, and the upper LOD is 300 AU/ml. The seropositivity threshold above which immunity by either vaccination or infection can be assumed is >16.5 AU/ml. Borderline titres (13.5–16.5 AU/ml) were considered as non-immune. For titres below the LOD half of the LOD was used for further analyses.

Due to the change in methods with different reference ranges, no titre levels were used for the analyses before liver transplantation, but children were divided into immune and non-immune. All measurements after transplantation were performed after 2015 on the Liaison indirect chemiluminescence immunoassay so that titre levels could be used for analysis.

### Statistical analysis

Discrete data is presented as numbers and percentages (%). Continuous data is given as median and the 25 to 75% quartile. Comparison of two groups with categorical variables was

done with the chi-square test or Fisher's exact test. For continuous variables in case of non-normality, either the Mann-Whitney U test was used or the Kruskal-Wallis test if there were more than two groups.

Cox proportional hazards regression analysis was performed to determine the association of a considered factor with measles immunity (positive measles IgG) after liver transplantation. The years between the last vaccination dose and test of measles immunity after liver transplantation were used as time interval. Hazard ratio (HR) and 95% confidence interval (CI) were reported. Variables with a P-value ≤ 0.200 were entered into a multivariable Cox proportional hazard regression. Linearity was evaluated using the Box-Tidwell procedure. The presence of multicollinearity was assessed by use of the correlation matrix, the tolerance and variance inflation factor.

One-way ANCOVA was conducted to determine differences between diagnoses and measles titre levels. For vaccinated children, the time from last vaccination to test was controlled for; for unvaccinated children, the time from liver transplantation to test was used. The linear relationship between the covariate and the outcome variable was assessed by visual inspection of a scatter plot for each group. Normality of the residuals was evaluated by visual inspection of the Q-Q plot. Homogeneity of variances was assessed using Levene's test. Homogeneity of regression slopes was analysed by checking the interaction term.

Statistical analysis was carried out with R version 4.2.2 [33]. For graphical data, the ggplot2 package version 3.4.1 [34], and for Cox proportional hazards regression analysis, the gtsummary package version 1.7.2 was used. P-values of <0.05 were considered statistically significant.

## Results

### Study population

For this study, 239 children and adolescents who were followed up at our institution between January 2021 and December 2022 with available vaccination records were identified. A total of 28 patients were excluded from analysis: 10 due to post-transplant lymphoproliferative disorder (PTLD) and 4 after hematopoietic stem cell transplantation. In addition, 14 received measles vaccination after liver transplantation and were analysed separately, see below. In the next step, those 211 patients were divided into vaccinated (n = 125) and unvaccinated (n = 86) pre transplantation on the basis of their vaccination record. Table 1 shows the characteristics of both groups classified by their measles IgG after transplantation as immune and non-immune.

In the group of children who did not receive measles vaccination before transplantation, there were no differences between the immune and non-immune groups for underlying disease group, gender, age at transplantation, graft type (whole organ or split), whether a living donation was performed, immunosuppression including trough level, and the time interval from transplantation to blood collection.

In the group of children vaccinated before transplantation, there is no significant influence of liver disease on measles immune status. However, patients with cadaveric grafts and whole grafts are more often immune to measles (p = 0.033 and p = 0.022). Measles-immune children are significantly older at transplantation, have a longer interval from both first and last vaccination to transplantation, received more vaccination doses and more are measles-seropositive at time of transplantation (each p<0.001). Interestingly, the interval from transplantation to testing is significantly shorter in the immune children (median 4.65 vs. 8.59 years, p = 0.002), although the two do not differ in age at testing (median 10.43 vs. 10.27 years, p = 0.96). In contrast, time from last vaccination to transplantation is significantly longer in the immune children (median 2.30 vs. 0.59 years, p<0.001), even though there is no difference in time from

**Table 1. Patient characteristics.** A classification of unvaccinated and vaccinated before transplantation was made. Both groups were then divided into immune and non-immune based on measles IgG after liver transplantation.

| | | Unvaccinated prior to transplantation | | | Vaccinated prior to transplantation | | |
|---|---|---|---|---|---|---|---|
| | | Measles immune (n = 23) | Measles non-immune (n = 63) | p | Measles immune (n = 94) | Measles non-immune (n = 31) | p |
| Diagnosis | | | | | | | |
| | Biliary/cholestatic | 21 (91.3%) | 41 (65.1%) | 0.12 | 54 (57.4%) | 15 (48.4%) | 0.077 |
| | Malignancy | 0 (0.0%) | 6 (9.5%) | | 8 (8.5%) | 8 (25.8%) | |
| | Acute liver failure | 1 (4.3%) | 2 (3.2%) | | 9 (9.6%) | 4 (12.9%) | |
| | Metabolic | 0 (0.0%) | 6 (9.5%) | | 18 (19.1%) | 2 (6.5%) | |
| | Cryptogenic | 1 (4.3%) | 8 (12.7%) | | 5 (5.3%) | 2 (6.5%) | |
| Sex | | | | | | | |
| | Female | 11 (47.8%) | 33 (52.4%) | 0.708 | 43 (45.7%) | 14 (45.2%) | 0.817 |
| | Male | 12 (52.2%) | 30 (47.6%) | | 51 (54.3%) | 17 (54.8%) | |
| Living related donor | | | | | | | |
| | Yes | 13 (56.5%) | 28 (44.4%) | 0.321 | 9 (9.6%) | 8 (25.8%) | 0.022 |
| | No | 10 (43.5%) | 35 (55.6%) | | 85 (90.4%) | 23 (74.2%) | |
| Whole graft | | | | | | | |
| | Yes | 4 (17.4%) | 10 (15.9%) | 0.866 | 38 (40.4%) | 6 (19.4%) | 0.033 |
| | No | 19 (82.6%) | 53 (84.1%) | | 56 (59.6%) | 25 (80.6%) | |
| Age at transplant, median (IQR) in years | | 0.57 (0.45–0.62) | 0.60 (0.49–0.84) | 0.197 | 4.53 (2.69–7.92) | 1.96 (1.15–3.28) | <0.001 |
| Number of vaccinations | | | | | | | |
| | 1 vaccination | n.a. | n.a. | n.a. | 15 (16.0%) | 16 (51.6%) | <0.001 |
| | 2 vaccinations | | | | 79 (84.0%) | 15 (48.4%) | |
| Age at 1st vaccination, median (IQR) in years | | n.a. | n.a. | n.a. | 1.03 (0.94–1.19) | 0.99 (0.86–1.10) | 0.111 |
| Time from first vaccination to transplantation, median (IQR) in years | | n.a. | n.a. | n.a. | 3.16 (1.34–6.64) | 0.68 (0.26–2.29) | <0.001 |
| Time from last vaccination to transplantation, median (IQR) in years | | n.a. | n.a. | n.a. | 2.30 (1.07–4.94) | 0.59 (0.16–1.58) | <0.001 |
| Time from last vaccination to testing, median (IQR) in years | | n.a. | n.a. | n.a. | 8.43 (5.43–11.61) | 9.35 (5.60–12.59) | 0.516 |
| Seroprevalence at time of transplantation | | | | | | | |
| | Yes | 0 (0.0%) | 0 (0.0%) | 1.000 | 85 (90.4%) | 17 (54.8%) | <0.001 |
| | No | 2 (100.0%) | 2 (100.0%) | | 9 (9.6%) | 14 (45.2%) | |
| Time from transplantation to testing, median (IQR) in years | | 9.39 (5.05–14.24) | 8.10 (2.22–12.66) | 0.342 | 4.65 (1.75–7.93) | 8.59 (4.33–10.53) | 0.002 |
| Age at testing, median (IQR) in years | | 9.76 (5.52–14.78) | 9.78 (2.76–14.05) | 0.465 | 10.43 (6.90–14.39) | 10.27 (5.98–13.76) | 0.960 |
| Biopsy proven acute rejection (RAI-Score≥3) | | | | | | | |
| | Yes | 7 (30.4%) | 22 (34.9%) | 0.697 | 36 (38.3%) | 13 (41.9%) | 0.719 |
| | No | 16 (69.6%) | 41 (65.1%) | | 58 (61.7%) | 18 (58.1%) | |
| Immunosuppression | | | | | | | |
| | Cyclosporine A | 7 (30.4%) | 21 (33.3%) | 0.800 | 9 (9.6%) | 5 (16.1%) | 0.316 |
| | Tacrolimus | 16 (69.6%) | 42 (66.7%) | | 85 (90.4%) | 26 (83.9%) | |
| Intensified IS | | | | | | | |

(*Continued*)

**Table 1.** (Continued)

| | | Unvaccinated prior to transplantation | | | Vaccinated prior to transplantation | | |
|---|---|---|---|---|---|---|---|
| | | Measles immune (n = 23) | Measles non-immune (n = 63) | p | Measles immune (n = 94) | Measles non-immune (n = 31) | p |
| | Yes | 2 (8.7%) | 10 (15.9%) | 0.395 | 22 (23.4%) | 13 (41.9%) | 0.046 |
| | No | 21 (91.3%) | 53 (84.1%) | | 72 (76.6%) | 18 (58.1%) | |
| Cyclosporine A trough level, median (IQR) [µg/l] | | 27.0 (27.0–44.0) | 40.0 (26.0–87.0) | 0.254 | 46.0 (35.5–61.5) | 35.0 (27.5–35.5) | 0.061 |
| Tacrolimus trough level, median (IQR) [µg/l] | | 2.6 (2.3–3.2) | 3.1 (2.4–3.9) | 0.230 | 3.3 (2.4–4.5) | 2.9 (2.5–3.4) | 0.276 |

last vaccination to testing (median 8.43 vs. 9.35 years, p = 0.516). There is no difference between the immune and non-immune groups with regard to immunosuppression used, as well as their trough levels. However, slightly significantly less intensified immunosuppression is common in the immune group (p = 0.046), even though there is no difference in the occurrence of biopsy proven acute rejections between both groups.

## Measles immunity at the time of and after liver transplantation in relation to the number of vaccinations performed

Of the 211 patients analysed, 31 (14.7%) were vaccinated once and 94 (44.5%) twice against measles before liver transplantation (Fig 1). Of the children vaccinated once, 22 (71.0%) were measles seropositive prior to transplant. However, this decreases following transplantation, so that only 15 of the 31 children (48.4%) still had detectable antibodies (p = 0.07). By contrast, 80

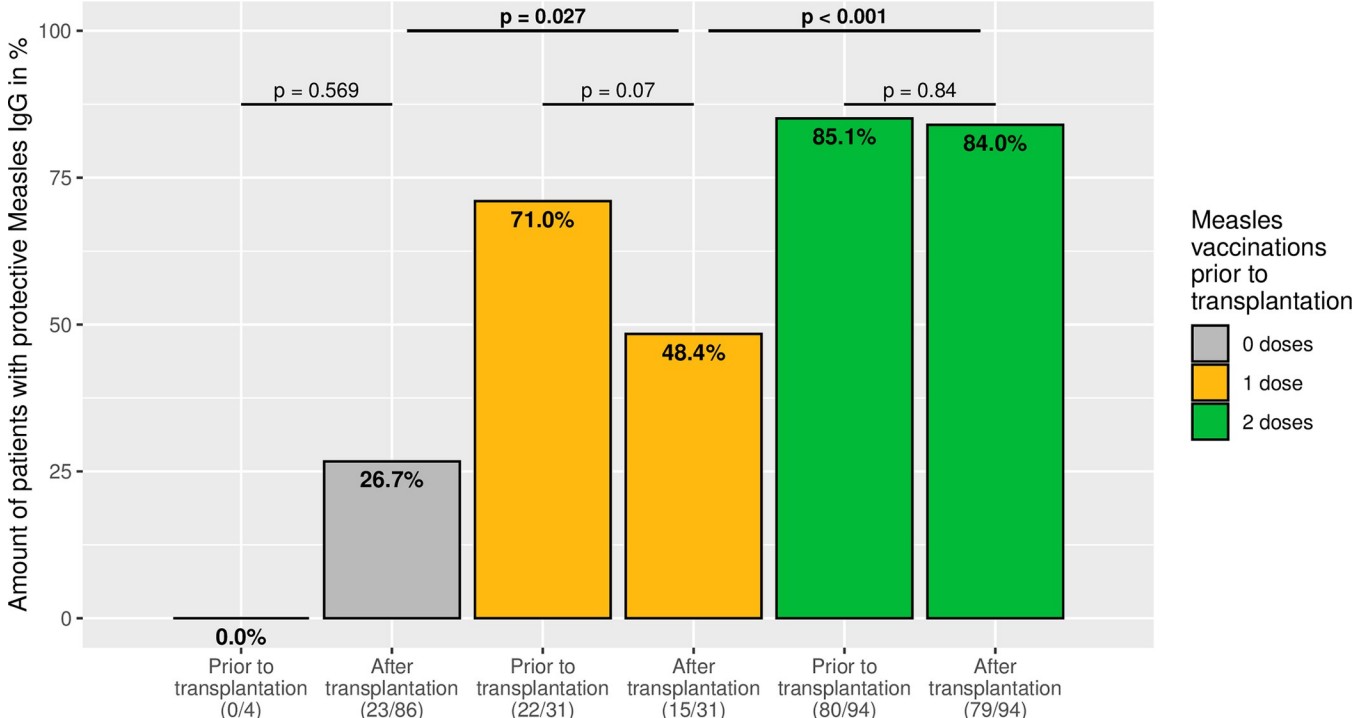

**Fig 1. Measles immunity in relation to the number of vaccinations performed.** Bar plots show the proportion of patients with a measles IgG titre indicating serological immunity at the time of and after transplantation, according to the number of measles vaccinations administered prior to transplantation.

of 94 (85.1%) of those vaccinated twice were measles immune before transplantation. This remained stable after transplantation, whereas 79 of the 94 (84.0%) were still seropositive (p = 0.84). Interestingly, in 23 of the 86 (26.7%) unvaccinated, non-immune children before transplantation, antibodies were detectable after transplantation, so that immunity could be assumed.

In summary, two measles vaccinations before liver transplantation show the best response, both before and after transplantation, without significant decrease. By contrast, after one vaccination, not even half of the children are seropositive after transplantation.

### Measles immunity in relation to the number of years from transplantation to testing and to the number of vaccinations administered

There may be a decrease in measles seropositivity over time after transplantation, therefore patients were divided into the following three time intervals, as shown in Fig 2, less than 5 years, between 5 and 10 years, and more than 10 years after transplantation. In order to minimise the influence of the number of vaccinations before transplantation, a subdivision was made on this basis. Particularly in the twice-vaccinated patients, there was a decline in measles IgG positive patients over time, but this was not significant (p = 0.167). There was no trend in the other groups. In summary, there is no significant loss of measles immunity over time following transplantation.

### Measles immunity in relation to the number of years from last vaccination to testing and change of seroprevalence after transplantation

Since an immune response is induced with each immunisation, the next step was to investigate measles immunity depending on the time of the last vaccination to testing and the number of

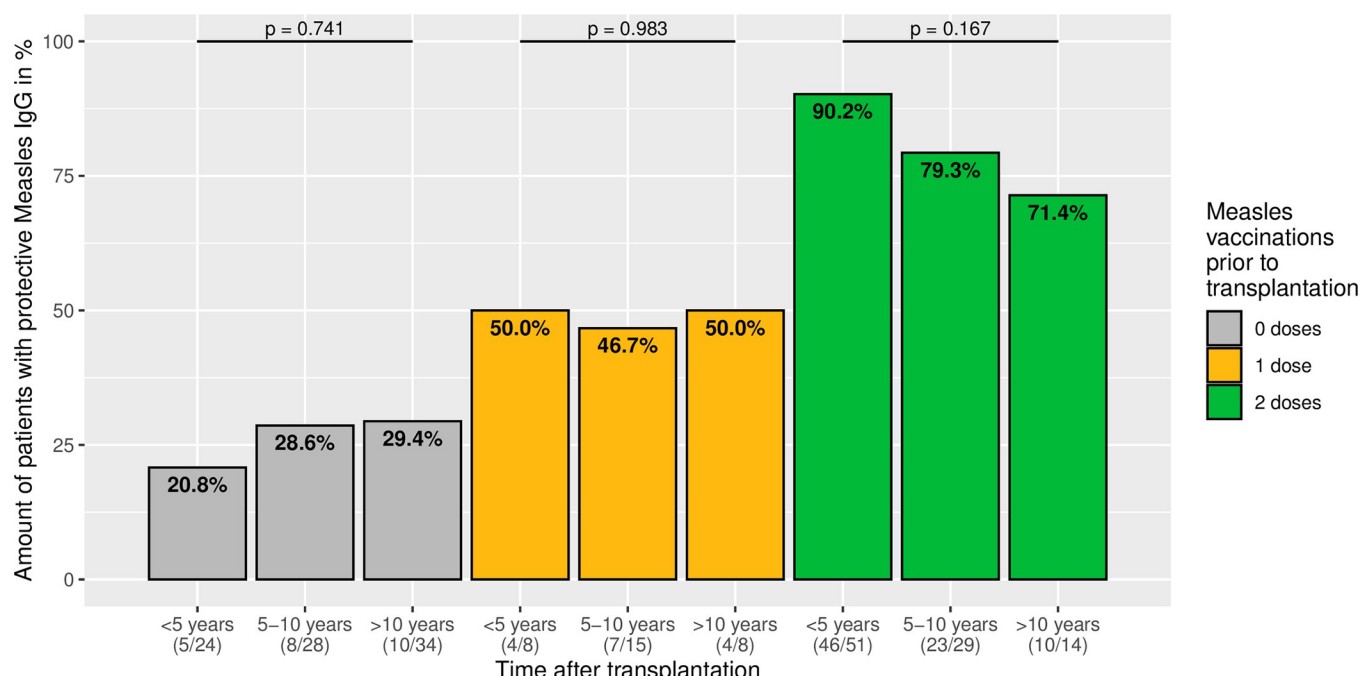

**Fig 2. Measles immunity in relation to the years from transplantation to testing.** The bar plots show the proportion of patients with a positive measles IgG after transplantation; according to the number of measles vaccinations administered prior to transplantation and in relation to time from transplantation. No significant trend was detectable.

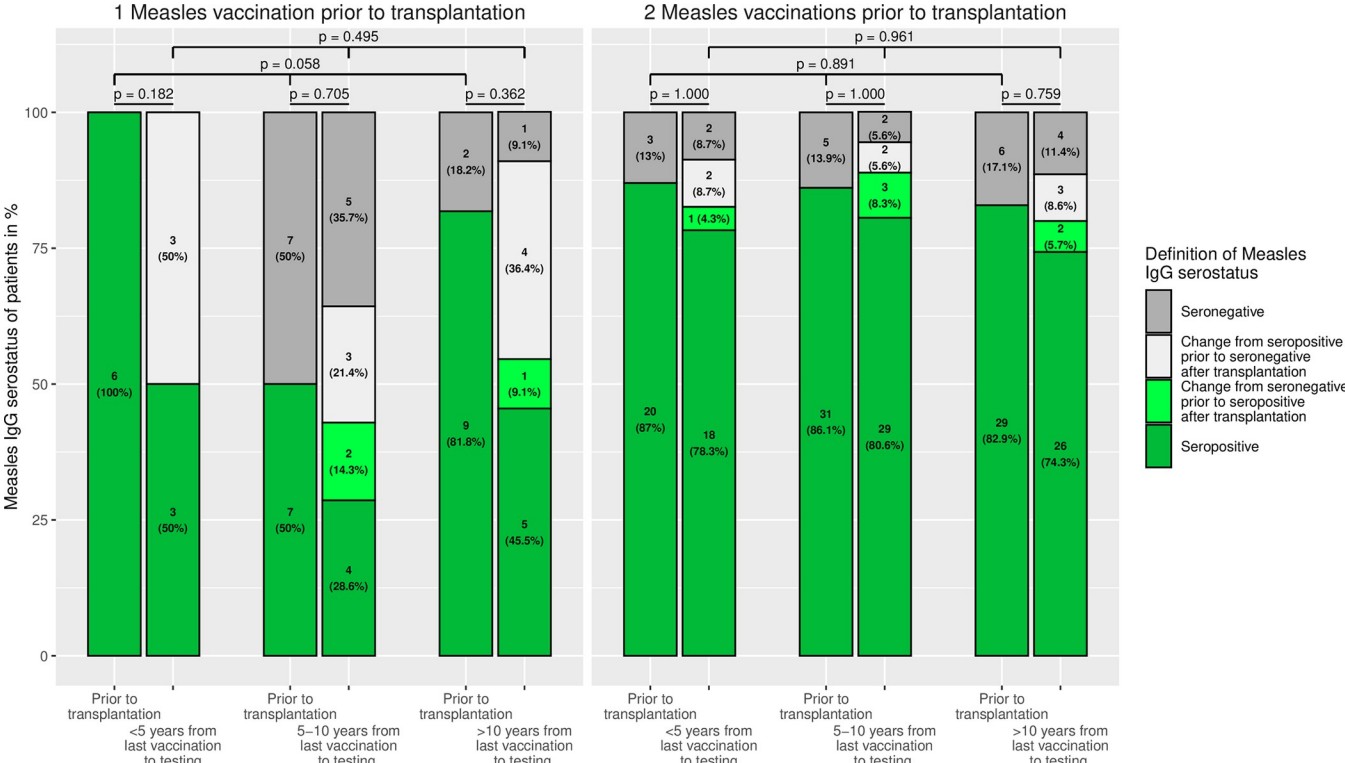

**Fig 3. Measles immunity in relation to the years from last vaccination to testing.** The bar plots show the measles serostatus of patients prior to and after transplantation; according to the number of measles vaccinations administered prior to transplantation and in relation to time from last vaccination to testing. No significant trend was detectable. However, about 5–15% of patients who are seronegative at the time of transplantation are seropositive afterwards.

vaccination doses (Fig 3). For this purpose, time intervals of less than 5 years, between 5 and 10 years, and more than 10 years after last immunisation were used. There is no significant difference from prior to after transplantation, nor with time interval after the last vaccination. The bar plots also show the change in measles serostatus from prior to after liver transplantation. In addition to the fact that some of the patients who were seropositive at transplantation showed a loss of measles immunity, about 5–15% developed antibodies after transplantation.

## Factors influencing the immunity of liver transplanted, measles vaccinated, paediatric patients

To investigate factors influencing measles immunity after paediatric liver transplantation, all vaccinated patients were classified on the basis of measles IgG after liver transplantation into immune (n = 94) and non-immune (n = 31). Table 2 shows the results of univariable and multivariable Cox proportional hazards regression of measles immunity with the time interval in years from the last vaccine dose to the test of measles immunity after liver transplantation. The following variables had a P-value ≤ 0.200 in univariable regression and were put into a multivariable Cox regression model: sex, seroprevalence at time of transplant, number of vaccinations pre transplant, age at first vaccination, living-related donor, immunosuppression, intensified immunosuppression, age at transplant and age at testing.

In the multivariable logistic regression model, we could identify four independent, significant associations to post-transplant measles immunity: the number of vaccinations performed pre-transplant (HR 4.30 [95% CI 2.09–8.83], $p < 0.001$), seroprevalence prior to

**Table 2. Univariable and multivariable Cox proportional hazards regression of measles immunity.**

| | Univariable Cox Regression | | | | Multivariable Cox Regression | | |
|---|---|---|---|---|---|---|---|
| | N | HR | 95% CI | p-value | HR | 95% CI | p-value |
| Diagnosis | | | | | | | |
| Metabolic disease | 20 | — | — | | | | |
| Biliary/cholestatic disease | 69 | 1.26 | 0.74, 2.17 | 0.394 | | | |
| Malignant liver disease | 16 | 1.21 | 0.52, 2.83 | 0.653 | | | |
| Acute liver failure | 13 | 0.66 | 0.30, 1.47 | 0.309 | | | |
| Cryptogenic cirrhosis/others | 7 | 1.03 | 0.38, 2.78 | 0.952 | | | |
| Sex | | | | | | | |
| Female | 57 | — | — | | — | — | |
| Male | 68 | 0.67 | 0.44, 1.01 | 0.058 | 1.24 | 0.76, 2.02 | 0.38 |
| Living related donation | | | | | | | |
| No | 108 | — | — | | — | — | |
| Yes | 17 | 0.63 | 0.32, 1.25 | 0.186 | 1.06 | 0.46, 2.43 | 0.90 |
| Type of graft | | | | | | | |
| Split graft | 81 | — | — | | | | |
| Whole graft | 44 | 1.13 | 0.75, 1.72 | 0.558 | | | |
| Age at transplantation in years | 125 | 0.94 | 0.89, 1.00 | 0.044 | 1.07 | 0.97, 1.18 | 0.19 |
| Vaccinations prior to transplantation | | | | | | | |
| 1 vaccine dose | 31 | — | — | | — | — | |
| 2 vaccines doses | 94 | 1.43 | 0.81, 2.51 | 0.200 | 4.30 | 2.09, 8.83 | <0.001 |
| Age at 1st vaccination in years | 125 | 1.16 | 0.99, 1.35 | 0.070 | 11.5 | 6.92, 19.1 | <0.001 |
| Age at 1st vaccination below 1 year | | | | | | | |
| No | 67 | — | — | | | | |
| Yes | 58 | 1.04 | 0.69, 1.57 | 0.854 | | | |
| Seroprevalence at time of transplantation | | | | | | | |
| No | 23 | — | — | | — | — | |
| Yes | 102 | 2.26 | 1.13, 4.50 | 0.021 | 2.38 | 1.07, 5.30 | 0.034 |
| Age at testing in years | 125 | 0.31 | 0.25, 0.40 | <0.001 | 0.09 | 0.06, 0.15 | <0.001 |
| Biopsy proven acute rejection (RAI-Score≥3) | | | | | | | |
| No | 76 | — | — | | | | |
| Yes | 49 | 0.98 | 0.64, 1.49 | 0.913 | | | |
| Immunosuppression | | | | | | | |
| Cyclosporine A | 14 | — | — | | — | — | |
| Tacrolimus | 111 | 1.63 | 0.79, 3.38 | 0.186 | 1.43 | 0.62, 3.32 | 0.40 |
| Intensified immunosuppression | | | | | | | |
| No | 101 | — | — | | — | — | |
| Yes | 24 | 1.52 | 0.94, 2.46 | 0.090 | 1.08 | 0.58, 2.00 | 0.81 |

transplantation (HR 2.38 [95% CI 1.07–5.30], p = 0.034) and higher age at time of first vaccination (HR 11.5 [95% CI 6.92–19.1], p < 0.001) are positively associated. In contrast, older age at testing is inversely associated with measles immunity (HR 0.09 [95% CI 0.06–0.15], p < 0.001).

## Diagnosis-dependent factors influencing the immunity of liver transplanted, measles vaccinated, paediatric patients

To investigate diagnosis-specific factors that may influence measles immunity, patients were subdivided according to their diagnosis group (biliary/cholestatic liver disease, malignant liver disease, acute liver failure, metabolic disease and cryptogenic cirrhosis/others) and measles

IgG (immune and non-immune) after transplantation. The results of each univariable and multivariable Cox proportional hazards regression of measles immunity with the time interval in years from the last vaccine dose to the test of measles immunity after liver transplantation are summarised in S1 Table.

Looking at the largest patient group (biliary/cholestatic disease), there was a positive association with seroprevalence prior to transplantation (HR 3.44 [95% CI 1.31–9.05], p = 0.012) and higher age at time of first vaccination (HR 6.81 [95% CI 3.80–12.2], p < 0.001), as in the overall cohort. Living related donation (HR 0.26 [95% CI 0.10–0.74] p = 0.011) and higher age at time of testing (HR 0.14 [95% CI 0.08–0.25], p < 0.001) are associated with poorer measles immunity.

The latter also has a negative association with measles immunity in patients with metabolic disease as the cause of liver transplantation (HR 0.40 [95% CI 0.24–0.66], p < 0.001). Other factors were not significant for this group.

For the patient groups malignant liver disease, acute liver failure and cryptogenic cirrhosis/others, no significant association could be detected in the multivariable Cox regression. Interestingly, none of the five separately analysed patient groups showed an association with the number of vaccination doses prior to transplantation.

## The underlying liver disease influences measles titre levels

Next, measles titres were examined. For this purpose, patients were again divided according to the number of vaccinations carried out before transplantation (0, 1 and 2). Based on their underlying disease (biliary/cholestatic liver disease, malignant liver disease, acute liver failure, metabolic disease and others), a subdivision was made. As shown in Fig 4, the titres increased with the number of vaccinations and was highest after two doses compared to one dose (p < 0.001).

One-way ANCOVA was run to determine the effect of different diagnoses on measles titre levels. For vaccinated children, the time from last vaccination to test was controlled for; for unvaccinated children, the time from liver transplantation to test was used. After adjustment for time, no impact of diagnosis on the titre levels was observed in the group of unvaccinated patients as well as in the group vaccinated once before transplantation ($F_{(4,76)}$ = 0.802, p = 0.527 and $F_{(2,25)}$ = 1.233, p = 0.308, respectively). In contrast, an influence of the diagnosis on measles titres after adjusting for time was found for twice-vaccinated children ($F_{(4,84)}$ = 3.475, p = 0.011). However, in post hoc analysis with Bonferroni adjustment there was only a significantly lower titre in children after acute liver failure compared to the those with underlying metabolic disease (p = 0.027).

## Measles immunisation after liver transplantation

The 14 children vaccinated against measles after transplantation and excluded from the above analyses are examined here in more detail. Half of the children were female (n = 7, 50%) and were transplanted at a median age of 0.89 (IQR: 0.53–1.4) years. Reasons for transplantation were biliary/cholestatic liver disease in 10 patients (71.43%), metabolic disease in 2 (14.29%), 1 patient with malignant liver disease and 1 with cryptogenic cirrhosis (7.14% each). A total of 8 children received the measles vaccination for the first time after transplantation: 4 children (28.6%) with one dose and 4 children (28.6%) with 2 doses. The remaining 6 children (42.9%) were vaccinated once before and received another dose after liver transplantation. Patients were a median of 4.30 (IQR: 3.02–10.57) years old at vaccination and thus a median of 3.12 (IQR: 2.00–10.06) years after liver transplantation. Measles IgG was measured at a median of 4.58 (IQR: 2.52–8.97) years after vaccination.

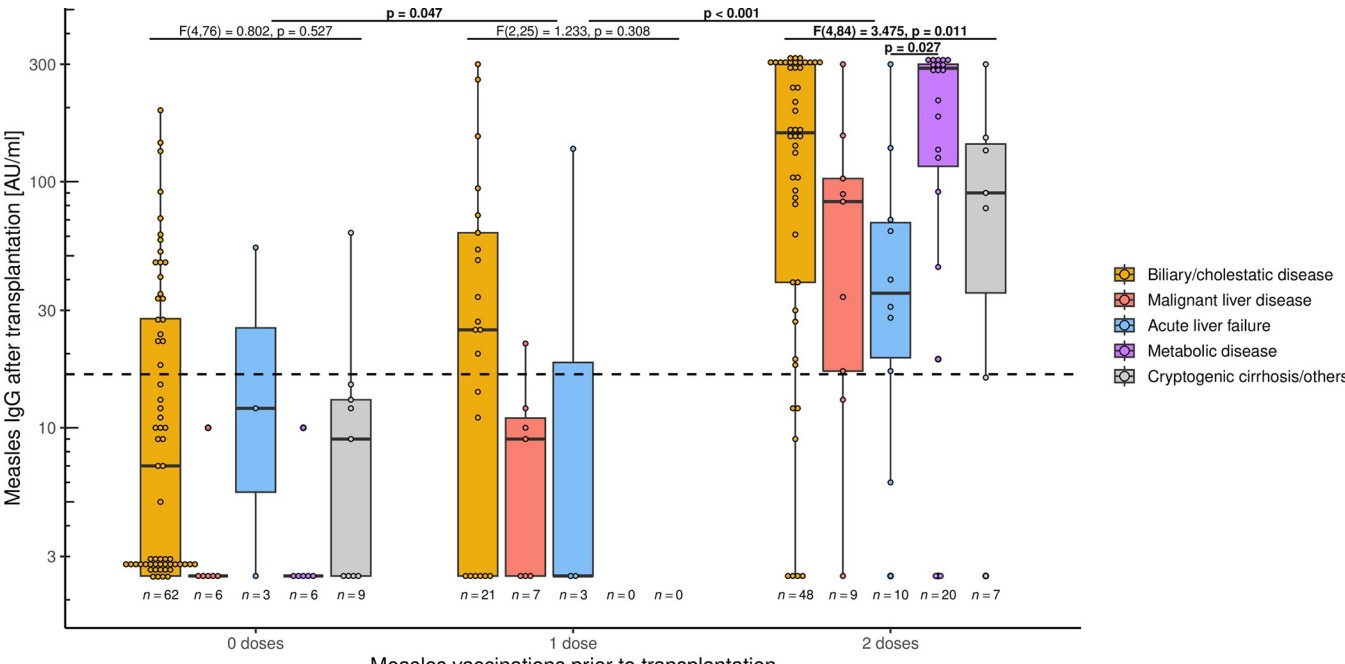

**Fig 4. Influence of underlying liver disease on measles titres.** Measles IgG titre after paediatric liver transplantation, depending on number of vaccinations prior to transplantation and underlying liver disease. Box plots show median and interquartile range. The y-axis is scaled logarithmically. The dashed line marks the threshold for serological protection. Measles titre is higher after two vaccinations than after one vaccination ($p < 0.001$). To determine the impact of diagnosis on measles titre levels, one-way ANCOVA analysis was conducted: After adjusting for time from transplantation to testing, there was no significant effect in unvaccinated children ($F_{(4,76)} = 0.802$, $p = 0.527$). Twice-vaccinated children, after adjusting for time from last vaccination to testing, showed a significant influence of diagnosis on measles titre ($F_{(4,84)} = 3.475$, $p = 0.011$), particularly between acute liver failure and underlying metabolic disease ($p = 0.022$). There was no effect in patients after one-off immunisations ($F_{(2,25)} = 1.233$, $p = 0.308$).

After one or two vaccination doses, 3 of the 4 (75%) patients were seropositive and 5 of the 6 (83.3%) after one vaccination each, before and after transplantation. There was no difference in the titre level ($p = 0.691$). In the retrospective survey, there was no association with rejections, nor did any child develop a measles infection.

## Discussion

Our study of 211 children and adolescents is one of the largest investigating measles immunity, including risk factors for non-immunity, after paediatric liver transplantation. More than 55% of all patients examined and 75% of those vaccinated at least once before transplantation had detectable antibodies indicating immunity. This is in line with a study by Liman et al. where measles IgG is present in almost 46% of all children following liver transplantation with and without vaccination beforehand [27]. Yoeli et al. were able to serologically detect measles immunity in 78% of those vaccinated at least once [29]. In contrast, Funaki et al. found that only 47% of children were protected [28]. However, in this study 90% of patients had only been vaccinated once, but this data matches our observation that about 48% have detectable measles antibodies after one vaccine dose. A study of 50 children with biliary atresia was able to detect antibodies in 84% after two immunisations—this before transplantation [26]. This is consistent with our results where about 85% of all children are measles IgG positive after two vaccinations. It should be noted, however, that over 95% of healthy children are already seropositive after one vaccination and thus the response is weaker in patients with liver disease [26, 35].

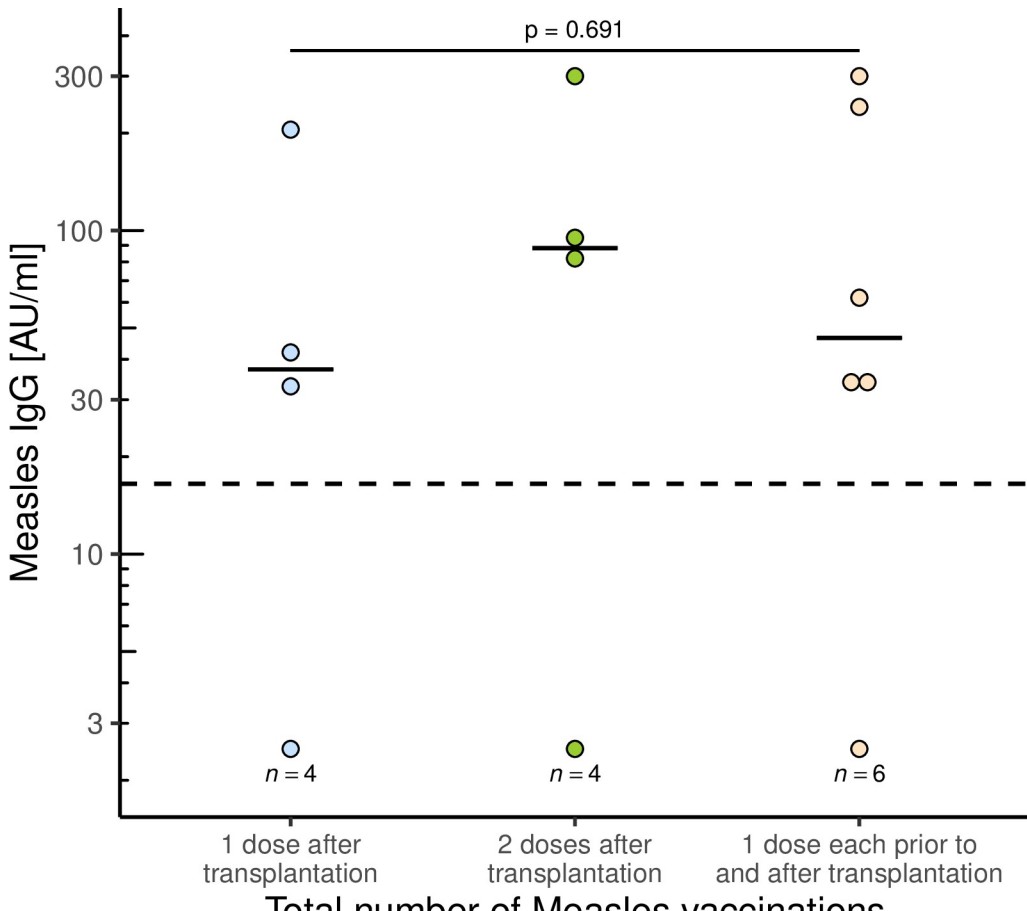

**Fig 5. Influence of measles vaccination after transplantation on measles titres.** A total of 14 children were vaccinated against measles after liver transplantation. Of these, 4 received one vaccination, 4 received two vaccinations and 6 received one vaccination both before and after transplantation. Seropositivity after vaccination was 75%, 75% and 83.3%, respectively, with no significant difference in titre between groups.

Interestingly, measles IgG is present in more than a quarter of our patients, even without vaccination. In order to exclude a false-positive measurement, a new measurement without change was taken at re-presentation. This group mainly includes patients who received a transplant at a very young age and thus did not attain the recommended age for live vaccinations [16, 17]. Moreover, 5–15% of children who were seronegative at the time of transplantation, despite previous vaccination, subsequently become seropositive (Fig 3). There are no characteristic differences between immune and non-immune patients without vaccination, although it should be noted that they are at median more than 8 years after liver transplantation. Similarly, Liman et al. were able to detect measles antibodies in 3 of 17 children without documented MMR vaccination [27]. This possibly indicates a post-transplant infection, even if this is not the case in any patients according to their medical history. It is striking that the titres are low, even if higher titres would be expected after an infection [36, 37]. T-cell mediated response plays a critical role in MMR vaccination [38] as well as in measles infection [39]. It is possible that the immune response and thus antibody production is weakened due to immunosuppression after transplantation, as all patients receive a calcineurin inhibitor (tacrolimus or cyclosporine A), which reduces T-cell function [40]. However, it remains unclear what is causative for the measles IgG detection. Due to immunosuppression, the measles rash may be

misdiagnosed as another infectious disease [41] or even be absent altogether [42]. Assuming an infection, the question arises whether vaccination could not be more liberal after transplantation. However, in a survey among centres of the Society of Pediatric Liver Transplantation (SPLIT) conducted by Kemme et al., only 24% check antibody titres and 29% offer live vaccinations after transplantation [43]. Actually, recent studies show good tolerability and immune response to MMR vaccination in children after liver transplantation [21, 22]. In our cohort, too, 14 children were vaccinated against measles after transplantation and, as far as can be assessed in a retrospective study, they tolerated the vaccine well and their measles titres responded (Fig 5). Perhaps the indication for vaccination after transplantation could be made more broadly if the patient's course is stable [23]. Although we cannot demonstrate a significant decline in measles immunity with time from transplantation (Fig 2), multivariable Cox regression showed that older age at testing is negatively associated with measles immunity, indicating that loss of immunity may occur over time. Thus, a booster vaccination could also be considered for children who were vaccinated before transplantation and lost immunity. A decision instrument for this would be to determine the serostatus regularly (e.g. every 5 years) in order to observe a trend.

Regarding risk factors for measles immunity: Yoeli et al. demonstrated, that a weight above 10kg at transplantation and a whole graft transplant are independently associated with measles immunity status [29]. In the study by Liman et al. younger age at transplantation as well as a longer interval from transplantation to testing are risk factors for non-immunity. However, seropositivity is associated with the number of documented vaccinations [27]. We could show that the number of vaccinations (HR 4.30 [95% CI 2.09–8.83], p<0.001), seropositivity before liver transplantation (HR 2.38 [95% CI 1.07–5.30], p = 0.034) and higher age at time of first vaccination (HR 11.5 [95% CI 6.92–19.1], p < 0.001) are independently associated with measles immunity after transplantation. In contrast, and consistent with the observation of Liman et al., older age at testing is inversely associated (HR 0.09 [95% CI 0.06–0.15], p < 0.001), indicating loss of immunity over time. As opposed to Yoeli et al., graft type has no significant influence on measles immunity, but in the subanalysis of patients with biliary/cholestatic disease, living related donation shows a poorer outcome (HR 0.26 [95% CI 0.10–0.74], p = 0.011). Due to organ shortage, this may offer an option for transplantation for younger or sicker patients in particular and explain why immunity is poorer [4, 44]. However, in the same study, PELD score at transplant, reflecting the medical urgency for transplantation, differed not between immune and non-immune patients. Importantly, and in contrast to the study by Funaki et al., vaccination below one year of age is not a significant factor for non-immunity [28]. Potentially, the observation of Funaki et al. is due to the high number of one-off immunisations with potential maternal antibodies. However, children for whom transplantation is anticipated in the first year of life, should receive live vaccines from the age of 6 months [23]. This may be the only way to develop an immune response prior to transplantation and increase vaccination coverage as well as seroprevalence beforehand.

A review from 2019 concluded that MMR vaccination under 9 months of age in healthy children also provides adequate protection, although effectiveness and antibody titres are higher when patients are older at the time of vaccination, resulting in a recommendation of MMR vaccination below 9 months of age for children at high risk of infection [45]. However, the median age for administration of both MMR vaccines in children with chronic liver disease is higher than the national average [19]. Early vaccination of high-risk patients, such as those with biliary atresia at 6 months of age, would at least increase the proportion of those immunised before transplantation. A small case series reports on the administration of live vaccines 7–21 days before SOT without any problems—although it has to be said that a total of 5 children were involved [46]. If the child's condition is sufficiently stable, further immunisation

can be done in due course. Moreover, as our study shows, testing whether these children are measles IgG positive is also helpful, as this is an independent factor for post-transplant immunity. Thus, children who have been vaccinated twice against MMR but are measles IgG negative could receive a third vaccination before transplantation to achieve measles immunity in the long term.

Measles immunity is relevant since a transient immunosuppression with increased susceptibility to infections has been described even in healthy children after infection [47]. Higher measles titres may protect against infection, but at least minimise disease severity, similar to other infectious diseases [48, 49]. Looking at the antibody titre levels, there is an increase, depending on the vaccine doses administered, just as in healthy people [35]. To assess the influence of the underlying disease on titre levels one-way ANCOVA was conducted: Only patients vaccinated twice before transplantation showed an effect after adjusting for time from last vaccination to testing, although in the post-hoc analysis this was only significant between groups with acute liver failure and underlying metabolic disease. The group with one-off immunisations, 31 patients, is probably too small to demonstrate significant effects—especially since no patients with metabolic disease or cryptogenic cirrhosis could be included. In addition, the measles titre in children without vaccination presumably results from infection, so that no differences are to be expected here.

Contrary to expectations and reports, children with malignant liver disease and after chemotherapy do not show the lowest titres [50, 51]. Interestingly, the titres in the group with acute liver failure are the lowest compared to metabolic disease. After all, the patients were healthy until the onset of liver failure and received their immunisations beforehand. Because of the sudden onset of acute liver failure, previous titres were rarely determined. Still, in half of the patients the reason for liver failure is unclear [52], but this patient group is prone to infections [53]. Since bone marrow failure occurs more frequently after acute liver failure [54], there are possible immunological effects behind it. Recent studies describe a form of activated CD8 T-cell hepatitis in children with indeterminate acute liver failure as an immunological condition [55]. In contrast, the spectrum of underlying metabolic diseases is heterogeneous, some of which are associated with immunodeficiency. In addition, hypoproteinaemia may be present due to protein aversion, for example in urea cycle disorders [56], but this is in contrast to the highest titres. In general, malnutrition may have little or no effect on the immune response to vaccination [57]. Unfortunately, there are no metabolic patients with only one vaccination to compare whether the effect can also be seen here. Future studies should therefore investigate the influence of the underlying disease on measles titres. Nevertheless, it should be kept in mind that patients potentially lose their vaccine antibodies more rapidly after acute liver failure and are therefore at risk of VPI.

Our study has its limitations: Firstly, it is a retrospective study. Due to its cross-sectional design, there are different intervals after transplantation, which makes comparison difficult. Secondly, we did not assess the potential impact of the different vaccines and whether coadministration makes a difference. However, no differences could be shown in healthy children regarding both points [35, 58]. Furthermore, we focused on measles IgG as a factor and determiner for cellular immunity against measles, but other factors may also play a part [59]. Larger studies are needed to further investigate the response to vaccination in this high-risk group in order to prevent vaccine-preventable infections.

## Conclusion

Measles non-immunity is frequent after paediatric liver transplantation. On the one hand, this is due to poorer seroconversion after vaccination compared to healthy children. On the other

hand, many children undergo transplantation without MMR vaccination because they do not attain the recommended age for live vaccines. Our data shows that both the number of vaccine doses and a pre-transplant positive measles IgG are independently associated with seropositivity after transplantation. As recent studies demonstrate good immunogenicity of MMR vaccines even below 9 months of age, liver transplant candidates should be vaccinated earlier, due to the increased risk of infection. In addition, a negative measles IgG, despite two vaccinations, also offers the possibility of a booster before transplantation to achieve long-term immunity.

The underlying liver disease, in combination with the number of vaccine doses, may influence the level of measles titre but not the seroprevalence, with patients after acute liver failure having the lowest titres. In addition, more than 26% of unvaccinated patients develop measles IgG after transplantation, as well as 5–15% of at time of transplant seronegative, vaccinated children are seropositive afterwards. Both possibly through infection. In contrast, older age at testing is negatively associated with measles immunity, indicating a loss of seroprevalence after transplantation. Recent studies show good tolerability of MMR vaccination even after transplantation. Thus, if the patient's course is stable, administration of live vaccinations after transplantation should also be discussed openly with the families. For this purpose, awareness of vaccinations as well as of the measurement of vaccine antibodies must also be raised among the treating transplantation specialists to reduce vaccine-preventable infections.

## Supporting information

**S1 Table. Univariable and multivariable Cox proportional hazards regression of measles immunity after vaccination for each diagnosis group analysed separately.**
(PDF)

## Acknowledgments

The authors would like to thank Angela Bain Emslie for help with editing.

## Author Contributions

**Conceptualization:** Tobias Laue, Ulrich Baumann.

**Data curation:** Tobias Laue.

**Formal analysis:** Tobias Laue, Norman Junge, Christoph Leiskau, Frauke Mutschler, Johanna Ohlendorf.

**Investigation:** Tobias Laue, Norman Junge.

**Validation:** Tobias Laue.

**Writing – original draft:** Tobias Laue, Ulrich Baumann.

**Writing – review & editing:** Tobias Laue, Norman Junge, Christoph Leiskau, Frauke Mutschler, Johanna Ohlendorf, Ulrich Baumann.

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
