## [Decision Letter · Decision Letter 0]

13 Sep 2023

PONE-D-23-25946Acute liver failure is associated with poorer measles immunity compared to other pre-transplant vaccinated children — A retrospective, single-centre, cross-sectional analysis after paediatric liver transplantationPLOS ONE

Dear Dr. Laue,

Thank you for submitting your manuscript to PLOS ONE. After careful consideration, we feel that it has merit but does not fully meet PLOS ONE’s publication criteria as it currently stands. Therefore, we invite you to submit a revised version of the manuscript that addresses the points raised during the review process.

We look forward to receiving your revised manuscript.

Kind regards,

Ashraf Elbahrawy

Academic Editor

PLOS ONE

Journal Requirements:

3. Thank you for stating the following in the Competing Interests section: "I have read the journal's policy and the authors of this manuscript have the following competing interests: Prof. U. Baumann is a consultant for Albireo Pharma, Mirum Pharma, Alexion Pharma, Vivet Pharma and Nestlé. Dr. C. Leiskau has received payments for a presentation by Albireo pharma".

6. Please amend either the abstract on the online submission form (via Edit Submission) or the abstract in the manuscript so that they are identical.

Additional Editor Comments:

Dear Dr. Tobias Laue

This manuscript reviewed by two independent reviewers , indeed some concerns need to be addressed before the manuscript can be considered for publication. Please review the reviewers comments

Reviewers' comments:

Reviewer's Responses to Questions

**Comments to the Author**

1. Is the manuscript technically sound, and do the data support the conclusions?

Reviewer #1: Yes

Reviewer #2: Partly

2. Has the statistical analysis been performed appropriately and rigorously? 

Reviewer #1: Yes

Reviewer #2: No

3. Have the authors made all data underlying the findings in their manuscript fully available?

Reviewer #1: Yes

Reviewer #2: No

4. Is the manuscript presented in an intelligible fashion and written in standard English?

Reviewer #1: Yes

Reviewer #2: Yes

5. Review Comments to the Author

Reviewer #1: Well written and addresses a gap

Reads well with good grammar and format

I have an issue with the title - it deceives the reader into thinking this manuscript is largely about acute liver failure (which just make up a small relevant finding) The manuscript is a retrospective study on measles immunity in paediatric liver transplant recipients in relation to pre-transplant vaccination. The findings are relevant and would inform measles vaccination practices in this group of patients.

Discussion - important aspects of earlier vaccination in pre-transplant patients is noted. It would also be important to note countries where measles outbreak have happened now have introduced earlier vaccinations for all paed patients which has been of benefit in patients who receive liver transplants at a young age(,1 yr of age)

Reviewer #2: Comments on Methodology and Results:

I have reviewed the manuscript titled "Acute Liver Failure is Associated with Poorer Measles Immunity Compared to Other Pretransplant Vaccinated Children—A Retrospective, Single-Centre, Cross-Sectional Analysis after Paediatric Liver Transplantation" submitted for publication in the PLOS ONE Journal. The study explores measles immunity in pediatric liver transplant recipients, a topic of significant clinical relevance. However, I have identified several concerns related to the methodology and the presentation and interpretation of results that need to be addressed before the manuscript can be considered for publication.

Methodological Concerns:

The authors described the methodology they used for determining the factors associated with measles immunity after transplant as follows: "A univariate logistic regression analysis was performed to determine the association of a considered factor with measles immunity (positive measles IgG) after liver transplantation. Estimated risks (odds ratio, OR) and 95% confidence interval (CI) were reported. Variables with a P‐ value ≤ 0.200 were entered into a multivariable logistic regression." I believe that "measles immunity after transplant" is a time-related event, and logistic regression analysis may not be the most appropriate statistical method to use, as it does not consider the time factor. I suggest that the authors repeat the analysis at different time points post-transplant using a Cox regression analysis model to better capture the dynamics of measles immunity over time.

Under the sub-title "Measles immunity in relation to the number of years from transplantation to testing and to the number of vaccinations administered," it is mentioned that there was a decline in measles seropositivity over time after transplantation. However, the manuscript does not provide the number of patients who experienced a loss of seropositivity. Figure 2, which displays the number of patients with protective measles IgG over different time periods according to the number of vaccination doses before transplant, might be misleading as it lacks a clear connection between the three columns in each group. To improve clarity, I suggest representing this data using lines for each patient over the different time periods. I think the idea is to show the changes of the protective measles IgG in each patient over time, so it should be linked together.

In Table 2, different disease categories were compared using cryptogenic liver cirrhosis as the reference group. This approach may not be appropriate because these disease etiologies are not directly comparable. I recommend that the authors analyze the impact of the presence or absence of each disease etiology separately on measles immunity after transplant to provide a more meaningful comparison.

Under the sub-title “The underlying liver disease influences measles titer levels”, it is not obvious which test was used for this comparison and there are other confounding factors not taking into consideration. I think it is inappropriate conclusion that needs to be revised.

Minor Comment:

There are several typographical and grammatical errors throughout the manuscript that require editing. For example, in Line 30, "the numbers of vaccinations" should be corrected to "the number of vaccinations."

6. PLOS authors have the option to publish the peer review history of their article (what does this mean?). If published, this will include your full peer review and any attached files.

Reviewer #1: No

Reviewer #2: No

---

## [Author Response · Author response to Decision Letter 0]

7 Nov 2023

Reviewer 1

I have an issue with the title - it deceives the reader into thinking this manuscript is largely about acute liver failure (which just make up a small relevant finding) The manuscript is a retrospective study on measles immunity in paediatric liver transplant recipients in relation to pre-transplant vaccination. The findings are relevant and would inform measles vaccination practices in this group of patients.

Thank you for your comment. We changed the title of the manuscript to “Diminished measles immunity after paediatric liver transplantation — A retrospective, single-centre, cross-sectional analysis”.

Discussion - important aspects of earlier vaccination in pre-transplant patients is noted. It would also be important to note countries where measles outbreak have happened now have introduced earlier vaccinations for all paed patients which has been of benefit in patients who receive liver transplants at a young age(,1 yr of age)

Thank you for your useful comment. We have added information on changes in vaccination policy due to the frequent measles outbreaks, e.g. by lowering the age for the first vaccination dose. However, there is little data on the impact on liver transplant candidates and recipients.

We hope that you will find the changes we made to the manuscript satisfactory, and that you will consider our manuscript for publication.

Reviewer 2:

I believe that "measles immunity after transplant" is a time-related event, and logistic regression analysis may not be the most appropriate statistical method to use, as it does not consider the time factor. I suggest that the authors repeat the analysis at different time points post-transplant using a Cox regression analysis model to better capture the dynamics of measles immunity over time.

Thank you for the important and helpful comment. As recommended, we have re-performed the calculations with a Cox proportional hazards regression analysis to better account for the time aspect in the analysis. In fact, the results have changed, in particular there is now also a significant influence of the time to testing, indicating a loss of immunity. However, this again underlines the fact that a measurement of measles IgG is helpful after transplantation and that a booster vaccination can be recommended if the patient's course is stable.

Figure 2, which displays the number of patients with protective measles IgG over different time periods according to the number of vaccination doses before transplant, might be misleading as it lacks a clear connection between the three columns in each group. To improve clarity, I suggest representing this data using lines for each patient over the different time periods. I think the idea is to show the changes of the protective measles IgG in each patient over time, so it should be linked together.

Thank you for your important comment. The figure serves to better demonstrate whether there is a loss of seroprevalence in vaccinated children over time or whether unvaccinated patients show an increase in seroprevalence (potentially due to cumulative risk for infection over time).

To show the changes in serostatus from prior to after liver transplantation, we have added an additional figure. This shows that about 5-15% of patients are seronegative at the time of transplant and then become seropositive, but also children lose their seroprevalence over time.

I recommend that the authors analyze the impact of the presence or absence of each disease etiology separately on measles immunity after transplant to provide a more meaningful comparison.

Thank you for your helpful comment. We performed a Cox proportional hazards regression analysis for each disease separately. For patients with biliary/cholestatic disease, living related donation is an additional independent risk factor for non-immunity after transplant.

In patients with metabolic disease, older age at testing also shows a negative impact on measles immunity. However, for the patient groups with malignant liver disease, acute liver failure and cryptogenic cirrhosis/others, no significant association could be detected in the multivariable Cox regression.

Under the sub-title “The underlying liver disease influences measles titer levels”, it is not obvious which test was used for this comparison and there are other confounding factors not taking into consideration. I think it is inappropriate conclusion that needs to be revised.

Thank you for your important comment. The data were re-analysed with a one-way ANCOVA to allow adjustment with covariates. The statistical method was described in the methods and the figure was revised.

An influence of the diagnosis on measles titres was only found for twice-vaccinated children. However, in post hoc analysis with Bonferroni adjustment there was only a significantly lower titre in children after acute liver failure compared to the those with underlying metabolic disease (p=0.027).

---

## [Decision Letter · Decision Letter 1]

18 Dec 2023

Diminished measles immunity after paediatric liver transplantation — A retrospective, single-centre, cross-sectional analysis

PONE-D-23-25946R1

Dear Dr. Laue,

We’re pleased to inform you that your manuscript has been judged scientifically suitable for publication and will be formally accepted for publication once it meets all outstanding technical requirements.

Kind regards,

Ashraf Elbahrawy

Academic Editor

PLOS ONE

Additional Editor Comments (optional):

Reviewers' comments:

Reviewer's Responses to Questions

**Comments to the Author**

1. If the authors have adequately addressed your comments raised in a previous round of review and you feel that this manuscript is now acceptable for publication, you may indicate that here to bypass the “Comments to the Author” section, enter your conflict of interest statement in the “Confidential to Editor” section, and submit your "Accept" recommendation.

Reviewer #1: All comments have been addressed

Reviewer #2: All comments have been addressed

2. Is the manuscript technically sound, and do the data support the conclusions?

Reviewer #1: Yes

Reviewer #2: Yes

3. Has the statistical analysis been performed appropriately and rigorously? 

Reviewer #1: Yes

Reviewer #2: Yes

4. Have the authors made all data underlying the findings in their manuscript fully available?

Reviewer #1: Yes

Reviewer #2: (No Response)

5. Is the manuscript presented in an intelligible fashion and written in standard English?

Reviewer #1: Yes

Reviewer #2: Yes

6. Review Comments to the Author

Reviewer #1: Revision is satisfactory - I have no further comments to add. Well written and well revised as well.

Reviewer #2: Many thanks for incorporating my suggestions in your revised manuscript. I would like to thank the authors for putting such effort.

7. PLOS authors have the option to publish the peer review history of their article (what does this mean?). If published, this will include your full peer review and any attached files.

Reviewer #1: No

Reviewer #2: No

---

## [Editor Report · Acceptance letter]

27 Jan 2024

PONE-D-23-25946R1 

PLOS ONE

Dear Dr. Laue, 

I'm pleased to inform you that your manuscript has been deemed suitable for publication in PLOS ONE. Congratulations! Your manuscript is now being handed over to our production team.

Kind regards, 

on behalf of

Prof. Ashraf Elbahrawy 

Academic Editor

PLOS ONE